# EventRPG: Event Data Augmentation with Relevance Propagation Guidance

**Mingyuan Sun[1,2], Donghao Zhang[3], Zongyuan Ge[4], Jiaxu Wang[1], Jia Li[5],
Zheng Fang[2,*] & Renjing Xu[1,*]**
[1]The Hong Kong University of Science and Technology (Guangzhou)  [2]Northeastern University
[3]Seeing Machines  [4]Monash University  [5]Peking University
mingyuansun20@gmail.com, peter.zhang1@seeingmachines.com,
zongyuan.ge@monash.edu, jwang457@connect.hkust-gz.edu.cn,
j.gaga.lee@gmail.com, fangzheng@mail.neu.edu.cn, renjingxu@hkust-gz.edu.cn

## Abstract

Event camera, a novel bio-inspired vision sensor, has drawn a lot of attention for its low latency, low power consumption, and high dynamic range. Currently, overfitting remains a critical problem in event-based classification tasks for Spiking Neural Network (SNN) due to its relatively weak spatial representation capability. Data augmentation is a simple but efficient method to alleviate overfitting and improve the generalization ability of neural networks, and saliency-based augmentation methods are proven to be effective in the image processing field. However, there is no approach available for extracting saliency maps from SNNs. Therefore, for the first time, we present Spiking Layer-Time-wise Relevance Propagation rule (`SLTRP`) and Spiking Layer-wise Relevance Propagation rule (`SLRP`) in order for SNN to generate stable and accurate CAMs and saliency maps. Based on this, we propose `EventRPG`, which leverages relevance propagation on the spiking neural network for more efficient augmentation. Our proposed method has been evaluated on several SNN structures, achieving state-of-the-art performance in object recognition tasks including N-Caltech101, CIFAR10-DVS, with accuracies of $85.62\%$ and $85.55\%$, as well as action recognition task SL-Animals with an accuracy of $91.59\%$. Our code is available at https://github.com/myuansun/EventRPG.

## 1 Introduction

With the advent of event cameras, researchers have focused on applying the brain-inspired technique to achieve a variety of tasks, as the asynchronous nature of event cameras mimics the way the biological visual system works (Gallego et al., 2020). Event cameras record the change in brightness of each pixel, and once the change in brightness of a pixel exceeds a predetermined threshold, an event is triggered. The intrinsic properties of event cameras give them several advantages over RGB cameras, including low power consumption, high dynamic range, low latency, and high temporal resolution. These benefits highlight the potential of event cameras in challenging scenarios, such as low-light and high-speed conditions, which has led to some research emphasizing the use of event cameras for robotic sensing in challenging situations (Zuo et al., 2022; Chen et al., 2023). Spiking Neural Network (SNN) (Maass, 1997) is a type of neural network that is inspired by the way biological neurons communicate with each other. By integrating the biological neuronal dynamics into individual neurons, SNN becomes capable of representing intricate spatio-temporal information and dealing with asynchronous data naturally, typically event-based data.

In terms of classification tasks, a number of event-based datasets, such as N-MNIST, N-Caltech101 (Orchard et al., 2015), and CIFAR10-DVS (Li et al., 2017), have been used to evaluate the performance of artificial neural networks (ANNs) and SNNs. However, the issue of overfitting still poses a significant challenge for event-based datasets. Data augmentation is an efficient method for improving the generalization and performance of a model. Lots of methods have been proposed to augment event-based data, for example, transferring classic geometric augmentations from image

---

*Corresponding authors.

field to event-based field (Li et al., 2022), randomly dropping events (Gu et al., 2021), and mixing two event streams with a randomly sampled mask (Shen et al., 2023). Nevertheless, current mixing augmentation strategies in event-based field do not consider the size and location information of label-related objects, and thus may produce events with incorrect labels and disrupt the training process. To address this problem in image processing field, Uddin et al. (2020); Kim et al. (2020) mix the label-related objects together based on the saliency information obtained from neural networks. This paradigm achieves better results compared with conventional non-saliency augmentations. Kim & Panda (2021b) was the first to explore acquiring saliency information from SNNs. In this work, Spiking Activation Map (SAM) was presented to reveal the model's attention by weighted adding of the intermediate feature maps. Since the CAMs obtained from this method are not related to the network's predictions, their precision regarding the shape and position of label-related objects is still not satisfactory.

In this paper, we present two novel methods extended from layer-wise relevance propagation (LRP) (Bach et al., 2015) to visualize the label-related saliency information of SNNs, each offering distinct advantages in terms of temporal precision and computational time. Moreover, guided by this saliency information, we develop two data augmentation approaches targeted at event-based data, demonstrating significant improvements in the performance and generalization capability of SNNs across multiple classification tasks.

Our contributions are summarized as follows:

- We propose Spiking Layer-Time-wise Relevance Propagation (`SLTRP`) and Spiking Layer-wise Relevance Propagation (`SLRP`) to accurately reveal the saliency information of SNNs. The former reveals information across time, while the latter is less time-consuming.

- We present RPGDrop and RPGMix. By dropping and mixing events with the guidance of relevance propagation obtained from `SLRP` or `STLRP`, the augmented samples exhibit increased diversity and tight correlation with the labels. Combined with several geometric data augmentations, we formulate our data augmentation strategy namely `EventRPG`.

- We evaluate our proposed saliency visualizing method and data augmentation method using various SNNs on event-based object and action recognition datasets. Experiments demonstrate that our `SLRP` and `SLTRP` can generate high quality CAMs and saliency maps with sub-optimal computing time. `EventRPG` achieves state-of-the-art performance on both object recognition and action recognition tasks with limited time consumption.

## 2 PRELIMINARY

Layer-wise Relevance propagation (LRP) was first introduced in (Bach et al., 2015) as a visualization tool for generating saliency maps that show the contribution of individual pixels in the input data to the model prediction or a specific class, facilitating the interpretability of neural networks. According to the LRP rule, we assign a value to each neuron to represent the neuron's contribution to the prediction or target class, called Relevance Score. The idea of LRP is to find a propagation rule satisfying the following definition.

**Definition 1** (Conservation Property). On a neural network with $L$ layers, the relevance score of each layer satisfies

$$c = \sum_i R_i^{(l)}, \qquad \forall l \in [0, L], \tag{1}$$

where $c$ is a constant value and $R_i^{(l)}$ is the relevance score of $i^{th}$ neuron on layer $l$. $R^{(0)}$ is the relevance score before input layer.

We calculate the relevance score of $i^{th}$ input neuron on layer $l$ $R_i^{(l-1)}$ by the sum of all relevance scores propagated from all the connections in this layer:

$$R_i^{(l-1)} = \sum_j R_{i \leftarrow j}^{(l-1,l)}. \tag{2}$$

Utilizing the $\alpha\beta$-rule (Montavon et al., 2017) and abandoning the bias term $b$, we obtain the relevance score that should be propagated from the $j^{th}$ neuron in layer $l$ to the $i^{th}$ neuron in layer $l - 1$:

$$R_{i \leftarrow j}^{(l-1,l)} = R_j^{(l)} \cdot \left( \alpha \cdot \frac{z_{ij}^+}{\sum_i z_{ij}^+} + \beta \cdot \frac{z_{ij}^-}{\sum_i z_{ij}^-} \right), \tag{3}$$

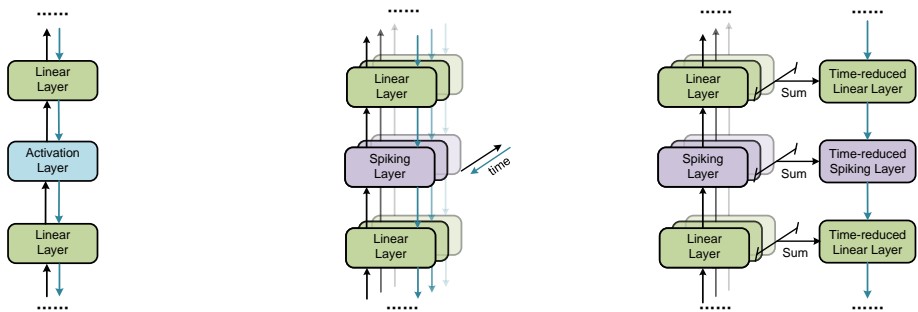

(a) Layer-wise Relevance Propagation    (b) Spiking Layer-Time-wise Relevance Propagation    (c) Spiking Layer-wise Relevance Propagation

Figure 1: Forward propagation flow and relevance propagation flow of ANNs (a) and SNNs (b, c).

where $\alpha + \beta = 1$. $z_{ij}^+ = w_{ij}^+ \cdot x_i^+ + w_{ij}^- \cdot x_i^-$ and $z_{ij}^- = w_{ij}^+ \cdot x_i^- + w_{ij}^- \cdot x_i^+$ are respectively the positive and negative contribution of the $i^{th}$ neuron in layer $l - 1$ to the $j^{th}$ neuron in layer $l$. This relevance propagation rule satisfies definition 1, and the proof can be seen in appendix D.1.

## 3   SPIKING RELEVANCE PROPAGATION RULE

In an SNN, information is represented by the timing and frequency of spikes. Neurons in an SNN generate spikes when their membrane voltage reaches a certain threshold. These spikes propagate through the network, influencing the activity of other neurons layer by layer. Two popular and fundamental models used to represent basic spiking neurons are the Leaky Integrate-and-Fire (LIF) neuron model and the Integrate-and-Fire (IF) neuron model:

$$f_{LIF}(V, I) = e^{-\frac{\Delta t}{\tau}} V[t-1] + \left(1 - e^{-\frac{\Delta t}{\tau}}\right) I[t], \tag{4}$$

$$f_{IF}(V, I) = V[t-1] + I[t], \tag{5}$$

where $\tau$ is the attenuation factor and $\Delta t$ is the interval between time steps satisfying $\Delta t << \tau$. Membrane voltage $V$ can be interpreted as the information reserved by a neuron from the previous time step $t - 1$. $I[t]$ indicates the input of the neuron at the current time step.

In practice, researchers would replace the activation layer in an ANN with spiking layers represented by eq. (4) or eq. (5) to construct an SNN. As shown in fig. 1a, an ANN consists of linear layers and activation layers and passes information layer by layer. These two layer types do not include variables related to time and as such we propagate the relevance scores through them layer by layer.

### 3.1   RELEVANCE PROPAGATION FOR LINEAR LAYERS IN SNNS

Information in SNNs is propagated through time in spiking layers, while the linear layers do not transmit information across different time steps. Any linear layer's output $y[t]$ at time step $t$ only correlates to its input $x[t]$ at time step $t$, and we conduct relevance propagation using eq. (3) on this layer separately for each time step. Specifically, for linear layer $l$, we extend the eq. (3) with suffix "$[t]$" representing the time step $t$, and we have

$$R_i^{(l-1)}[t] = \sum_j R_{i \leftarrow j}^{(l-1,l)}[t], \tag{6}$$

$$R_{i \leftarrow j}^{(l-1,l)}[t] = R_j^{(l)}[t] \cdot \left(\alpha \cdot \frac{z_{ij}^+[t]}{\sum_i z_{ij}^+[t]} + \beta \cdot \frac{z_{ij}^-[t]}{\sum_i z_{ij}^-[t]}\right), \tag{7}$$

in which $z_{ij}^+[t] = w_{ij}^+ \cdot x_i^+[t] + w_{ij}^- \cdot x_i^-[t]$ and $z_{ij}^-[t] = w_{ij}^+ \cdot x_i^-[t] + w_{ij}^- \cdot x_i^+[t]$. Next, we focus on the derivation of the relevance propagation rule on spiking layers.

### 3.2   SPIKING LAYER-TIME-WISE RELEVANCE PROPAGATION

From eq. (4) we know that in the forward propagation, the output of a LIF neuron at time step t ($t > 0$) depends on this layer's input current $I[t]$ at the current time step and the membrane voltage $V[t - 1]$ from the previous time step. Therefore, we should propagate the relevance score to the neuron's internal voltage at the previous time step and to the input current at the current time step, as shown in fig. 1b. Assume there are overall T time steps. Similar to $\alpha\beta$-rule, we propagate

the relevance score based on its positive contribution and negative contribution at each time step $t \in [1, T]$. Consider a neuron in spiking layer $l$ at time step t. We decomposite and represent it in a general manner:

$$f(V, I) = c \cdot V[t-1] + d \cdot I[t] \tag{8}$$

where c and d are coefficients depending on the neuron type:

$$c = \begin{cases} e^{-\frac{\Delta t}{\tau}} & \text{LIF Neuron,} \\ 1 & \text{IF Neuron,} \end{cases} \qquad d = \begin{cases} 1 - e^{-\frac{\Delta t}{\tau}} & \text{LIF Neuron,} \\ 1 & \text{IF Neuron.} \end{cases}$$

Based on eq. (8), at time step t, we define the proportion of relevance score that should be propagated to the previous time step as

$$\gamma[t] = \alpha \cdot \frac{c \cdot V^+[t-1]}{c \cdot V^+[t-1] + d \cdot I^+[t]} + \beta \cdot \frac{c \cdot V^-[t-1]}{c \cdot V^-[t-1] + d \cdot I^-[t]}, \tag{9}$$

where superscripts $^+$ and $^-$ denote the positive and negative values, respectively. For each neuron in spiking layer $l$, By initializing the relevance score at the final time step as $R^{(l-1)}[T] = R^{(l)}[T]$ and subsequently updating relevance scores iteratively as time step $t$ from $T$ to 1 using

$$R^{(l-1)}[t-1] \leftarrow \gamma[t] \cdot R^{(l-1)}[t] + R^{(l)}[t-1], \tag{10}$$

$$R^{(l-1)}[t] \leftarrow (1 - \gamma[t]) \cdot R^{(l-1)}[t], \tag{11}$$

we propagate the relevance score of every neuron in spiking layers to all time steps. With iteration formulas (10) and (11), we could derive the relevance score at each time step

$$R^{(l-1)}[t] = (1 - \gamma[t]) \left( \sum_{i=t+1}^{T} R^{(l)}[i] \prod_{j=t+1}^{i} \gamma[j] + R^{(l)}[t] \right). \tag{12}$$

**Proposition 1.** The sum of relevance scores propagated after spiking layer $l$ from time step 1 to $k(k < T)$ is

$$\sum_{t=1}^{k} R^{(l-1)}[t] = \sum_{t=1}^{k} R^{(l)}[t] + \sum_{i=k+1}^{T} R^{(l)}[i] \prod_{j=k+1}^{i} \gamma[j]. \tag{13}$$

The proof is provided in appendix D.2. With proposition 1, we have

$$\sum_{t=1}^{T} R^{(l-1)}[t] = \sum_{t=1}^{T-1} R^{(l-1)}[t] + R^{(l-1)}[T] = \sum_{t=1}^{T-1} R^{(l)}[t] + \gamma[T]R^{(l)}[T] + (1 - \gamma[T])R^{(l)}[T]$$

$$= \sum_{t=1}^{T-1} R^{(l)}[t] + R^{(l)}[T] = \sum_{t=1}^{T} R^{(l)}[t]. \tag{14}$$

This is a stronger Conservation Property since the relevance score stays unchanged for every neuron in a spiking layer. This indicates that leveraging formulas (10) and (11), we are able to propagate relevance scores through any spiking layer while satisfying the Conservation Property. Combining with relevance propagation rules introduced in section 3.1, we could propagate relevance scores to any layer in an SNN, thereby revealing the saliency information across time, namely Spiking Layer-Time-wise Relevance Propagation (SLTRP).

### 3.3 Spiking Layer-wise Relevance Propagation

SLTRP could reveal the saliency information at any time step, and would thus cost more time to conduct the whole relevance propagation process compared with ANNs. Under some circumstances, e.g., on datasets transformed from static images where coordinates of events tend to be fixed w.r.t. the time, we don't need to obtain the saliency score of some specific time step and only require the saliency information stacked from all time steps $R^{(l)} \equiv \frac{1}{T} \sum_{t}^{T} R^{(l)}[t]$. For any spiking layer $l$, from eq. (14) we have $R^{(l-1)} = R^{(l)}$.

In terms of linear layers, we sum the positive and negative contribution values of the $i^{th}$ neuron to $j^{th}$ neuron through time dimension as

$$z_{ij}^+ \equiv \frac{1}{T} \sum_{t} z_{ij}^+[t] = \frac{1}{T} \sum_{t} (w_{ij}^+ \cdot x_i^+[t] + w_{ij}^- \cdot x_i^-[t]) = \frac{1}{T} (w_{ij}^+ \cdot \sum_{t} x_i^+[t] + w_{ij}^- \cdot \sum_{t} x_i^-[t]),$$

$$z_{ij}^- \equiv \frac{1}{T} \sum_{t} z_{ij}^-[t] = \frac{1}{T} \sum_{t} (w_{ij}^+ \cdot x_i^-[t] + w_{ij}^- \cdot x_i^+[t]) = \frac{1}{T} (w_{ij}^+ \cdot \sum_{t} x_i^-[t] + w_{ij}^- \cdot \sum_{t} x_i^+[t]).$$

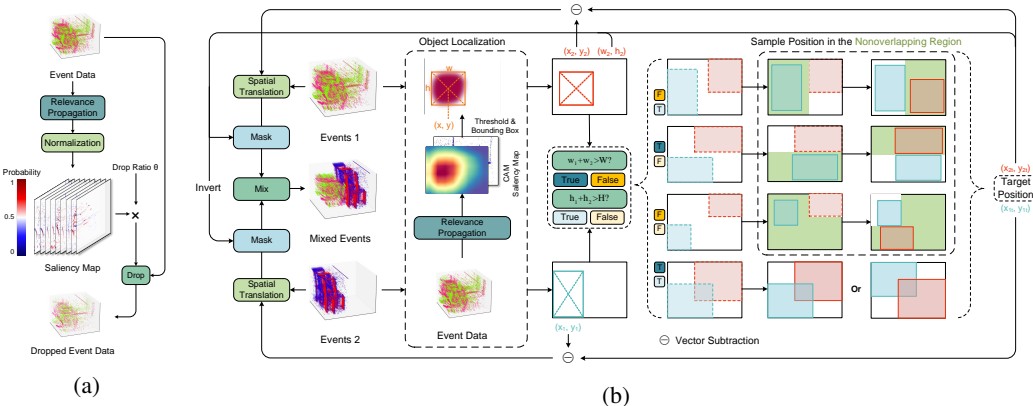

(a)                (b)

Figure 2: Illustration of Relevance Propagation Guided Event Mix and Event Drop. **(a) RPGDrop**. Where the saliency map offers a higher value, events are more likely to be dropped.**(b) RPGMix**.

Then we propagate the relevance scores of linear layer using eq. (2) and eq. (3) with different definitions of $R^{(l-1)}$, $R^{(l)}$, $z_{ij}^+$ and $z_{ij}^-$. This enables a spiking relevance propagation process without consideration of the time dimension, saving time costs and being more practical for datasets transformed from static images, namely Spiking Layer-wise Relevance Propagation (SLRP) (see fig. 1c).

## 4 RELEVANCE PROPAGATION GUIDED EVENT DATA AUGMENTATION

### 4.1 SALIENCY MAP AND CLASS ACTIVATION MAP

In an SNN, we first leverage Contrastive Layer-wise Relevance Propagation (CLRP) (Gu et al., 2018) to initialize the relevance scores of the output layer for each time step. Then we propagate the relevance scores backward using SLRP or SLTRP depending on the dataset. Relevance scores are propagated throughout all layers to create saliency maps, while class activation maps (CAMs) can be formed in two ways. One method involves summing the relevance scores from a specific intermediate layer across the channel dimension, resulting in SLRP-CAM and SLTRP-CAM. Alternatively, CAMs can be generated by calculating a weighted sum between relevance scores and feature maps, a method referred to as SLRP-RelCAM and SLTRP-RelCAM (Lee et al., 2021).

### 4.2 RELEVANCE PROPAGATION GUIDED EVENT DROP

Gu et al. (2021) has proven randomly dropping events to be an effective augmentation strategy. Furthermore, we expect to drop events more frequently in regions with label-related objects, motivated by the fact that disturbing regions with no label-related information (namely background) would have a negligible impact on the classifier's prediction. The label-related information can be provided by CAM and saliency map, where higher values imply higher model attention and label relevance. Since saliency map accurately reveals the relevance score of each pixel to the target in the input data, we leverage saliency map to guide dropping, detailly illustrated in fig. 2a. The higher the value of a pixel, the higher the probability that we will drop events on that pixel. $\theta$ is the parameter controlling the magnitude of augmentation.

### 4.3 RELEVANCE PROPAGATION GUIDED EVENT MIX

Event-based data, in contrast to image-based data, does not include color details, with the most crucial aspect being the texture information it contains. The overlapping of label-related objects will impair the texture details of these objects, which in turn further degrades the quality of features extracted in SNNs. Building upon this motivation, we propose Relevance Propagation Guided Event Mix (RPGMix). The whole mixing strategy is illustrated in fig. 2b. For two event-based data candidates, we utilize relevance propagation to localize the label-related regions and obtain two bounding boxes. To mix two objects with clear texture features, we randomly select two positions ensuring minimal overlap of their bounding boxes. This involves initially positioning one box at a corner to maximize the nonoverlapping area for the other box's placement, then selecting positions for both boxes in order, maintaining minimal overlap and maximizing sampling options. Finally, the two event streams are moved to the sampled positions. Although this linear translation part prevents the overlapping of label-related objects, the background of one object would still overlap with the other object. Moreover, in one single time step, the representation ability of the spiking neurons

(which only output binary information) is much worse than that of the activation layer (usually ReLU) of ANNs, making them less capable of spatial resolution and more likely to fall into local optima. Therefore, to promise the presence of only events from a single event stream candidate per pixel, avoiding regions with mixed information from interfering with the SNN, we adopt a CutMix strategy to mask the two event streams based on the bounding box of the second event stream, as demonstrated in the left part in fig. 2b. Kim et al. (2020) takes the sum of each sample's mask as the ratio of their corresponding labels. This ensures that the proportion of labels in the mixed label matches the proportion of pixels belonging to each sample. In our approach, we further aim to align the proportion of labels with the proportion of label-related pixels, which can be estimated using the bounding boxes. As a result, the labels of the two event streams are mixed as

$$L_{mix} = \frac{L_1(w_1h_1 - S_{overlap}) + L_2w_2h_2}{w_1h_1 + w_2h_2 - S_{overlap}}, \tag{15}$$

where $w_i$ and $h_i$ denote the width and height of the bounding box in the event stream $i$. $L_1$ and $L_2$ are the one-hot labels of the two event streams and $S_{overlap}$ is the area of the overlapping region of the two bounding boxes.

### 4.4 EVENTRPG

Combined with a few geometric data augmentation methods in NDA (Li et al., 2022), we formulate our data augmentation strategy namely `EventRPG` as shown in fig. 3. Specifically, for a batch of input event streams, each of the event streams would be augmented with randomly sampled policy and magnitude. They then have a probability of $0.5$ to be augmented by RPGMix.

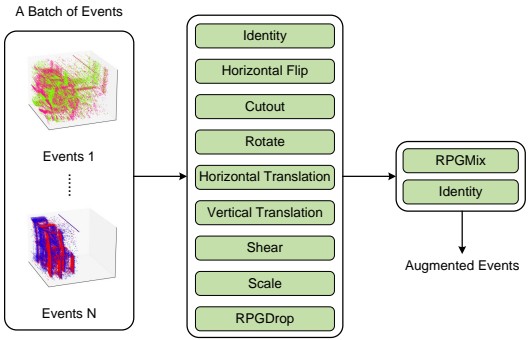

Figure 3: Augmentation process of `EventRPG`.

## 5 EXPERIMENTS

### 5.1 EFFECTIVENESS OF SLRP AND SLTRP

In this subsection, we evaluate the effectiveness of our approach for generating CAMs and saliency maps. Current event-based datasets for classification can be divided into two tasks: object recognition task and action recognition task. Generally, the former mainly involves event streams generated from jittering of static images, while the latter mainly involves event streams recorded in real environments, containing more dynamic information. We perform experiments on both types of datasets to showcase the effectiveness of our approach. We visualize the feature before the last fully connected layer as CAM.

| Method | Object Recognition | | | | | | Action Recognition | | | |
|---|---|---|---|---|---|---|---|---|---|---|
| | N-Caltech101 | | CIFAR10-DVS | | N-Cars | | DVSGesture | | SL-Animals | |
| | A.I. ↑ | A.D. ↓ | A.I. ↑ | A.D. ↓ | A.I. ↑ | A.D. ↓ | A.I. ↑ | A.D. ↓ | A.I. ↑ | A.D. ↓ |
| SAM (Kim & Panda, 2021b) | 0.86 | 22.33 | 1.23 | 46.83 | 5.24 | 5.89 | 8.27 | 10.83 | 20.60 | **8.23** |
| Grad-CAM (Selvaraju et al., 2017) | 12.03 | 41.71 | 8.31 | 22.15 | 21.77 | 17.74 | 0.41 | 67.09 | 0.80 | 81.44 |
| Grad-CAM++ (Chattopadhay et al., 2018) | 24.82 | 10.32 | 6.23 | 26.33 | **26.04** | **5.41** | 7.99 | 10.41 | 11.78 | 22.21 |
| SLRP-RelCAM (Lee et al., 2021) | 17.51 | 10.44 | 7.60 | 23.41 | 22.05 | 19.00 | 13.11 | 6.41 | **26.49** | 10.67 |
| SLTRP-RelCAM (Lee et al., 2021) | 17.54 | 10.44 | 7.60 | 23.41 | 22.05 | 19.00 | 13.05 | **6.40** | **26.49** | 10.67 |
| SLRP-CAM | **34.24** | 5.75 | 8.41 | 23.50 | 15.72 | 23.80 | 7.79 | 29.99 | 10.88 | 40.95 |
| SLTRP-CAM | **34.24** | 5.75 | 8.41 | 23.50 | 15.72 | 23.80 | 7.79 | 30.02 | 10.88 | 40.95 |
| SLRP-Saliency Map | 34.12 | **4.18** | 9.44 | 19.99 | 22.98 | 6.77 | **22.20** | 13.86 | 22.30 | 20.57 |
| SLTRP-Saliency Map | 34.17 | 4.19 | **9.51** | **19.98** | 22.84 | 6.75 | 21.17 | 13.95 | 22.30 | 20.79 |

Table 1: Comparison of A.I. and A.D. on event-based object recognition and action recognition datasets. We highlight the best results in bold and the second best results with underlining.

#### 5.1.1 OBJECTIVE FAITHFULNESS

We adopt two widely used metrics, Average Drop (A.D.) and Average Increase (A.I.), to measure the objective faithfulness of our method compared with other typical visualization tools. These two metrics explain how well an attention map explains a model's attention by measuring the average change in the model's output when the attention map is applied as a mask to the input. The higher the A.I. and the lower the A.D., the better the attention map explains the model's attention. We

compare our method with Grad-CAM (Selvaraju et al., 2017), Grad-CAM++ (Chattopadhay et al., 2018), and SAM (Kim & Panda, 2021b) on both object recognition datasets and action recognition datasets.

On object recognition task, our method achieves the best performance in terms of A.I. and A.D. on large datasets including N-Caltech101 and CIFAR10DVS, demonstrating the effectiveness of our method in generating CAMs and saliency maps. On the N-Cars dataset, Grad-CAM++ outperforms our method. This disparity may be due to N-Cars being a binary classification task that solely focuses on detecting the presence of a car in the event stream, in contrast to other datasets requiring multi-class object localization.

In terms of action recognition task, methods with spiking relevance propagation outperform other methods significantly, with an A.I. metric almost thrice as good as the best method without spiking relevance propagation — Grad-CAM++, in the DVSGesture dataset. In the SL-Animals dataset, RelCAM achieves the best results in terms of A.I. and also has a low A.D. value. Note that RelCAM is also obtained based on the relevance scores from `SLRP` and `SLTRP`. Therefore, its good results also helps to demonstrate the effectiveness of our spiking relevance propagation rules.

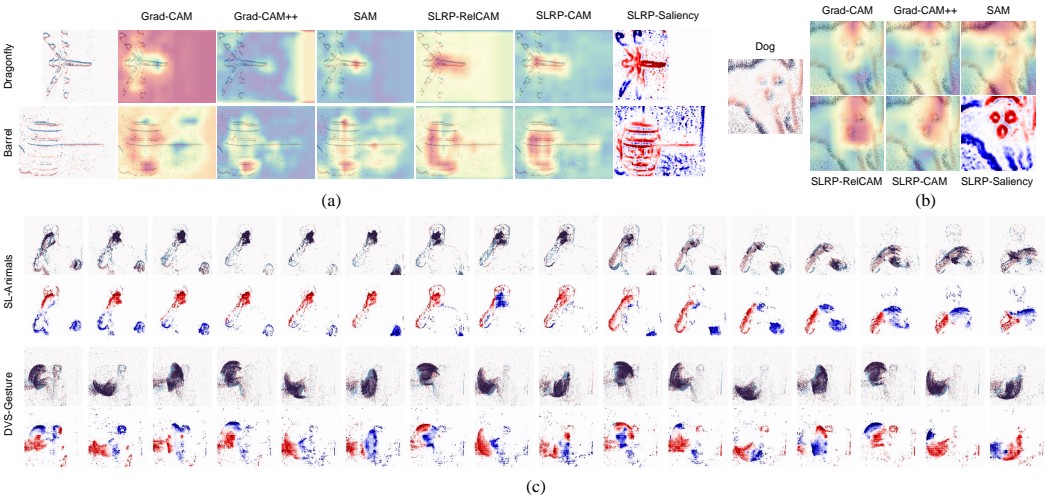

Figure 4: CAM and saliency map by different methods on **(a)** N-Caltech101 and **(b)** CIFAR10-DVS. **(c)** Saliency maps generated from `SLTRP` on DVS-Gesture and SL-Animals.

### 5.1.2 EVALUATION OF SELECTIVITY

We visualized the saliency map and CAM results of ours and other methods from Spiking-VGG11 and SEW Resnet18. As shown in fig. 4a, our method is more selective than other methods, with a higher value on the label-related objects and a lower value on the background. In contrast, Grad-CAM and SAM are more likely to be affected by the background, and Grad-CAM++ failed to locate the "dragonfly" on the top row, even though there are no other label-unrelated events in this sample. On CIFAR10-DVS, SLRP-CAM and SLRP-RelCAM successfully localize the dog's head, whereas the attention of the other methods drifted to other regions.

The saliency maps generated from `SLTRP` are able to track the exact moving object in the dataset (see fig. 4c). In the demonstration of SL-Animals dataset, the saliency map first focuses on the hand part to recognize the gesture. In the later time steps, it transfers its attention to the arm to capture the moving information. This proves its capability of temporal saliency information capturing , also yielding high selectivity.

| Model | Resolution | Grad-CAM | Grad-CAM++ | SLRP-RelCAM | SLTRP-RelCAM | SAM | SLRP-CAM | SLRP-Saliency Map | SLTRP-CAM | SLTRP-Saliency Map |
|---|---|---|---|---|---|---|---|---|---|---|
| Spiking VGG-11 | (48, 48) | 0.0776 | 0.0884 | **0.0225** | **0.0233** | 0.0267 | 0.0246 | 0.0645 | **0.0279** | 0.1157 |
| | (128, 128) | 0.0842 | 0.0996 | **0.0335** | **0.0347** | 0.0303 | 0.0323 | 0.0716 | **0.0345** | 0.1886 |
| SEW Resnet18 | (128, 128) | 0.2629 | 0.2846 | **0.0902** | **0.1072** | 0.0909 | 0.0869 | 0.1960 | **0.0785** | 0.2881 |

Table 2: Average time cost (s) of generating CAM and saliency map on N-Caltech101 (Spiking-VGG11) and DVSGesture (SEW Resnet18). Results within the quickest tier are highlighted in bold.

### 5.1.3 COMPUTATION TIME

As shown in table 2, SLRP-CAM, SLTRP-CAM are on the same level as SLTRP-RelCAM, SLRP-RelCAM, and SAM, all among the fastest methods. SLRP-Saliency Map and SLTRP-Saliency Map are slower than other methods, while still being competitive on SEW Resnet18 compared with Grad-CAM and Grad-CAM++. The time cost of SLTRP-CAM does not increase a lot compared to SLRP-CAM, since it only requires the relevance computation of the last fully connected layer. In contrast, the time cost of SLTRP-Saliency Map is much higher than SLRP-Saliency Map, since it requires the relevance computation of all layers.

## 5.2 RESULTS OF EVENTRPG

| Dataset | Data Augmentation | Training Method | Neural Network | Neuron | Timesteps | Resolution | Accuracy |
|---|---|---|---|---|---|---|---|
| N-Caltech101 | Flip | SALT (Kim & Panda, 2021a) | Spike-VGG16 | LIF | 20 | (80,80) | 55.00 |
| | NDA (Li et al., 2022) | STBP-tdBN (Zheng et al., 2021) | Spike-VGG11 | LIF | 10 | (48,48) | 78.20 |
| | NDA (Li et al., 2022) | STBP-tdBN (Zheng et al., 2021) | Spike-VGG11 | LIF | 10 | (128,128) | 83.70 |
| | Eventmix (Shen et al., 2023) | STBP | Pre-Act Resnet18 | PLIF | 10 | (48, 48) | 79.47 |
| | Identity | | | | | | 75.70 |
| | EventDrop (Gu et al., 2021) | TET (Deng et al., 2022) | Spike-VGG11 | LIF | 10 | (128,128) | 74.04 |
| | EventRPG (CAM) | | | | | | 85.00 |
| | EventRPG (Saliency Map) | | | | | | **85.62** |
| CIFAR10-DVS | Drop by time | STBP | SEW Wide-7B-Net | PLIF | 16 | (128,128) | 74.40 |
| | Flip | SALT (Kim & Panda, 2021a) | Spike-VGG16 | LIF | 20 | (64,64) | 67.10 |
| | Random Crop | DSR (Meng et al., 2022) | Spike-VGG11 | IF | 20 | (48,48) | 75.03 ± 0.39 |
| | Random Crop | DSR (Meng et al., 2022) | Spike-VGG11 | LIF | 20 | (48,48) | 77.27±0.24 |
| | FlipTranslation | TET (Deng et al., 2022) | Spike-VGG11 | LIF | 10 | (48,48) | 83.17±0.15 |
| | NDA (Li et al., 2022) | STBP-tdBN (Zheng et al., 2021) | Spike-VGG11 | LIF | 10 | (48,48) | 79.60 |
| | NDA (Li et al., 2022) | STBP-tdBN (Zheng et al., 2021) | Spike-VGG11 | LIF | 10 | (128,128) | 81.70 |
| | Eventmix (Shen et al., 2023) | STBP | Pre-Act Resnet18 | PLIF | 10 | (48,48) | 81.45 |
| | Identity | | | | | | 78.85 |
| | EventDrop (Gu et al., 2021) | TET (Deng et al., 2022) | Spike-VGG11 | LIF | 10 | (48, 48) | 77.73 |
| | EventRPG (CAM) | | | | | | **85.55** |
| | EventRPG (Saliency Map) | | | | | | 84.96 |
| N-Cars | NDA (Li et al., 2022) | STBP-tdBN (Zheng et al., 2021) | Spike-VGG11 | LIF | 10 | (48,48) | 90.10 |
| | NDA (Li et al., 2022) | STBP-tdBN (Zheng et al., 2021) | Spike-VGG11 | LIF | 10 | (128,128) | 91.90 |
| | Eventmix (Shen et al., 2023) | STBP | Pre-Act Resnet18 | PLIF | 10 | (48,48) | **96.29** |
| | Identity | | | | | | 94.92 |
| | EventDrop (Gu et al., 2021) | TET (Deng et al., 2022) | Spike-VGG11 | LIF | 10 | (48,48) | 95.46 |
| | EventRPG (CAM) | | | | | | 95.76 |
| | EventRPG (Saliency Map) | | | | | | 96.00 |

Table 3: Accuracy of various data augmentation methods on event-based object recognition datasets. All of the datasets used are created using event cameras. Specifically, N-Caltech101 and CI-FAR10DVS are derived from static images, while N-Cars is recorded in real-world environments.

In this section, we evaluate our proposed `EventRPG` with other augmentation methods including EventDrop (Gu et al., 2021), NDA (Li et al., 2022), and EventMix (Shen et al., 2023) across several object recognition and action recognition datasets. They could illustrate the performance of our methods in terms of static objects and moving objects, respectively. The datasets used and training settings are introduced in appendix C in details. Since `SLRTP` and `SLRP` yield nearly identical results for object recognition tasks, we only leverage `SLRP` in object recognition experiments since it costs fewer time. Eventdrop (Gu et al., 2021) did not conduct experiments on SNNs, so we reproduce it using its public code and conduct experiments under the same training setting of ours. We report the best accuracy for each experiment using same random seed.

### 5.2.1 OBJECT RECOGNITION TASKS

From table 3 we see that `EventRPG` achieves state-of-the-art performance on N-Caltech101 and CIFAR10-DVS datasets, bringing 9.92% and 6.7% improvements compared with identity (no augmentation), respectively. On N-Cars, `EventRPG` achieves the second-best performance, only 0.29% lower than EventMix. This might be attributed to the fact that N-Cars is a binary classification dataset, which only contains label "car" and "background". Most samples belonging to "background" do not have a specific label-related object to locate, making it difficult for our method to generate saliency map and CAM with high quality, thus decreasing the performance.

### 5.2.2 ACTION RECOGNITION TASKS

We implement `EventRPG` on SEW Resnet18 for action recognition tasks. On DVSGesture dataset, our method achieves best results compared with other reproduced augmentation methods under the same training settings, though slightly lower than EventMix on Pre-Act Resnet18. On SL-Animals dataset, our method achieves state-of-the-art performance among all data augmentation apporaches

| Model | Spike | Method | DVSGesture | SL-Animals | |
|-------|-------|--------|------------|------------|---|
| | | | | 4 sets | 3 sets |
| 7-Layer Spiking CNN | Hybrid | SCTFA (Cai et al., 2023) | **98.96** | **90.04** | - |
| GoogLeNet | ✗ | TORE (Baldwin et al., 2022) | 96.20 | 85.10 | - |
| Event Transformer | ✗ | EvT (Sabater et al., 2022) | 96.20 | 88.12 | **87.45** |
| Pre-Act Resnet18 | ✓ | EventMix (Shen et al., 2023) | 96.75 | - | - |
| SEW Resnet18 | ✓ | Identity | 94.33 | 85.42 | 89.09 |
| | | EventDrop (Gu et al., 2021) | 92.33 | 86.33 | 88.99 |
| | | NDA (Li et al., 2022) | 93.67 | 87.77 | 89.55 |
| | | EventRPG (CAM, SLRP) | 95.83 | 90.97 | 91.96 |
| | | EventRPG (Saliency Map, SLRP) | 95.49 | **91.59** | 93.30 |
| | | EventRPG (CAM, SLTRP) | 96.18 | 90.54 | 90.63 |
| | | EventRPG (Saliency Map, SLTRP) | **96.53** | 90.04 | **93.75** |

Table 4: Accuracy of various data augmentation methods on event-based action recognition datasets.

and all neural networks, with $3.82\%$ and $4.2\%$ improvements compared with the second-best augmentation method on 4 sets and 3 sets, respectively. Action recognition tasks include more dynamic information compared to object recognition tasks. Thus, the success of our method on action recognition tasks demonstrates its great potential for other dynamic event-based datasets, which is a future direction worth exploring.

### 5.2.3 TIME CONSUMPTION ANALYSIS

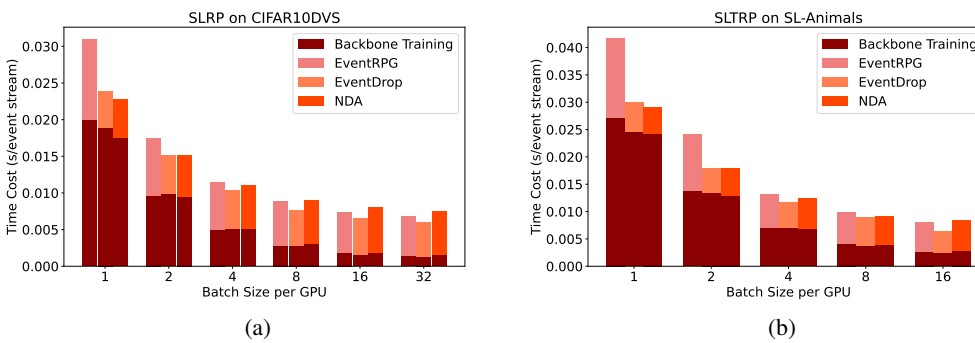

Figure 5: Average time cost of augmentation methods with SEW Resnet18 on two datasets.

In order to assess the computational efficiency of our data augmentation method, we perform two experiments to measure the time required for executing `EventRPG` with `SLRP` and `SLTRP` in object recognition and action recognition tasks. As depicted in fig. 5, the computation time of other data augmentation approaches remains constant regardless of the batch size, whereas our method exhibits a nearly linear decrease in computation time as the batch size increases, since the relevance propagation process, which is the main contributor to the time consumption, can be speeded up by parallizing computing across samples in a batch, similar to the gradient backpropagation process. When the batch size for each GPU exceeds 4, both `SLRP` and `SLTRP` achieve comparable speeds to NDA and EventDrop, confirming the time efficiency of `EventRPG`.

## 6 CONCLUSION AND LIMITATION

**Conclusion** In this paper, for the first time, we propose `SLTRP` and `SLRP`, two efficient and practical methods for generating CAMs and saliency maps for SNNs. Building upon this, we propose `EventRPG`, i.e., dropping events and mixing events with Relevance Propagation Guidance. Since `EventRPG` only disturbs and mixes regions on which model concerns most, it is more efficient compared to vanilla dropping and mixing, and also alleviates the likely misalignment problem between data and label. In our experiments, `SLRP` and `SLTRP` not only both yeild best results compared with other feature visualization tools, but also consume very little time to compute. `EventRPG` achieves state-of-the-art performance on N-Caltech101, CIFAR10-DVS and SL-Animals datasets, proving its strong generalization ability across different models and datasets.

**Limitation** Currently, `EventRPG` can only be implemented on classification tasks, remaining as a limitation. However, we could still leverage multi-task training paradim which has been proven to be effective to implement it into other downstream tasks, and we will also explore more possibilities of using `EventRPG` in self-supervised learning tasks.

## 7 ACKNOWLEDGEMENT

We would like to thank all anonymous reviewers for their committed work and insightful feedback. This work was supported in part by the National Natural Science Foundation of China under Grants 62073066 and U20A20197, in part by the Fundamental Research Funds for the Central Universities under Grant N2226001, in part by 111 Project under Grant B16009, in part by the Intel Neuromorphic Research Community (INRC) Grant Award (RV2.137.Fang), and in part by Guangzhou-HKUST(GZ) Joint Funding Program under Grant 2023A03J0682.

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

# APPENDIX

## A  ADDITIONAL EXPERIMENTAL RESULTS

Here we present more experiments to further prove the effectiveness of our approach.

### A.1  CLASSIFICATION ON MINI N-IMAGENET DATASET

N-ImageNet, as proposed by Kim et al. (2021), represents the neuromorphic adaptation of the well-known ImageNet dataset. It encompasses a thousand object categories, making it the most challenging task in event-based classification to date. We conduct experients to evaluate the performances of different data augmentation methods on mini N-ImageNet with SEW Resnet-18.

| Data Augmentation | Identity | EventDrop | NDA | EventRPG |
|---|---|---|---|---|
| Top-1 Accuracy | 28.16 | 34.18 | 35.84 | **40.90** |
| Top-5 Accuracy | 52.14 | 60.94 | 63.64 | **67.74** |

Table 5: Top-1 and Top-5 Accuracies (%) on mini N-ImageNet.

Table 5 demonstrates that data augmentations significantly enhance the performance of the model, as shown by the notable positive effects. In this comparison, our approach outperforms the nearest competitor NDA, achieving a higher Top-1 accuracy by $5.06\%$ and Top-5 accuracy by $4.1\%$.

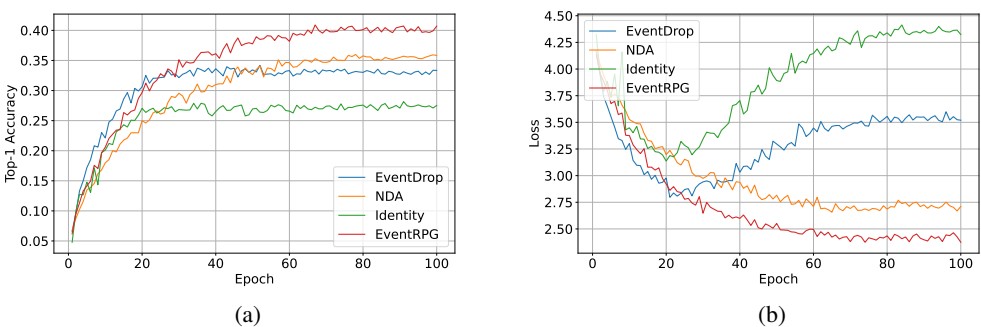

(a)                (b)

Figure 6:  The loss and Accuracy curves of different augmentations on mini N-ImageNet.

As shown in fig. 6a, the performance disparity between `EventRPG` and NDA is as significant as the one observed between EventDrop and Identity. Additionally, late epochs of EventDrop and Identity exhibit overfitting, a problem that is effectively mitigated in NDA and `EventRPG`, as illustrated in fig. 6b. These outcomes highlight again the efficiency of our approach in boosting the model's performance and mitigating overfitting.

### A.2  COMPARISON WITH OTHER SALIENCY-BASED MIX METHODS

We also compare the performance of various saliency-based mix methods on two typical datasets: N-Caltech101 and SL-Animals. In the experients, there is a $50\%$ chance for each event stream to be augmented by corresponding mix approach with no other augmentations applied. Notably, it's observed in table 6 that both Saliency Mix (Uddin et al., 2020) and Puzzle Mix (Kim et al., 2020) occasionally result in lower performance than the identity approach. This underscores RPGMix's robust generalization capability, as it consistently excels across all datasets.

To clearly showcase the differences among these mix methods, we present visualizations of the augmented samples they produce (see fig. 7). In most cases, our RPGMix aims to preserve as many label-related pixels as possible.

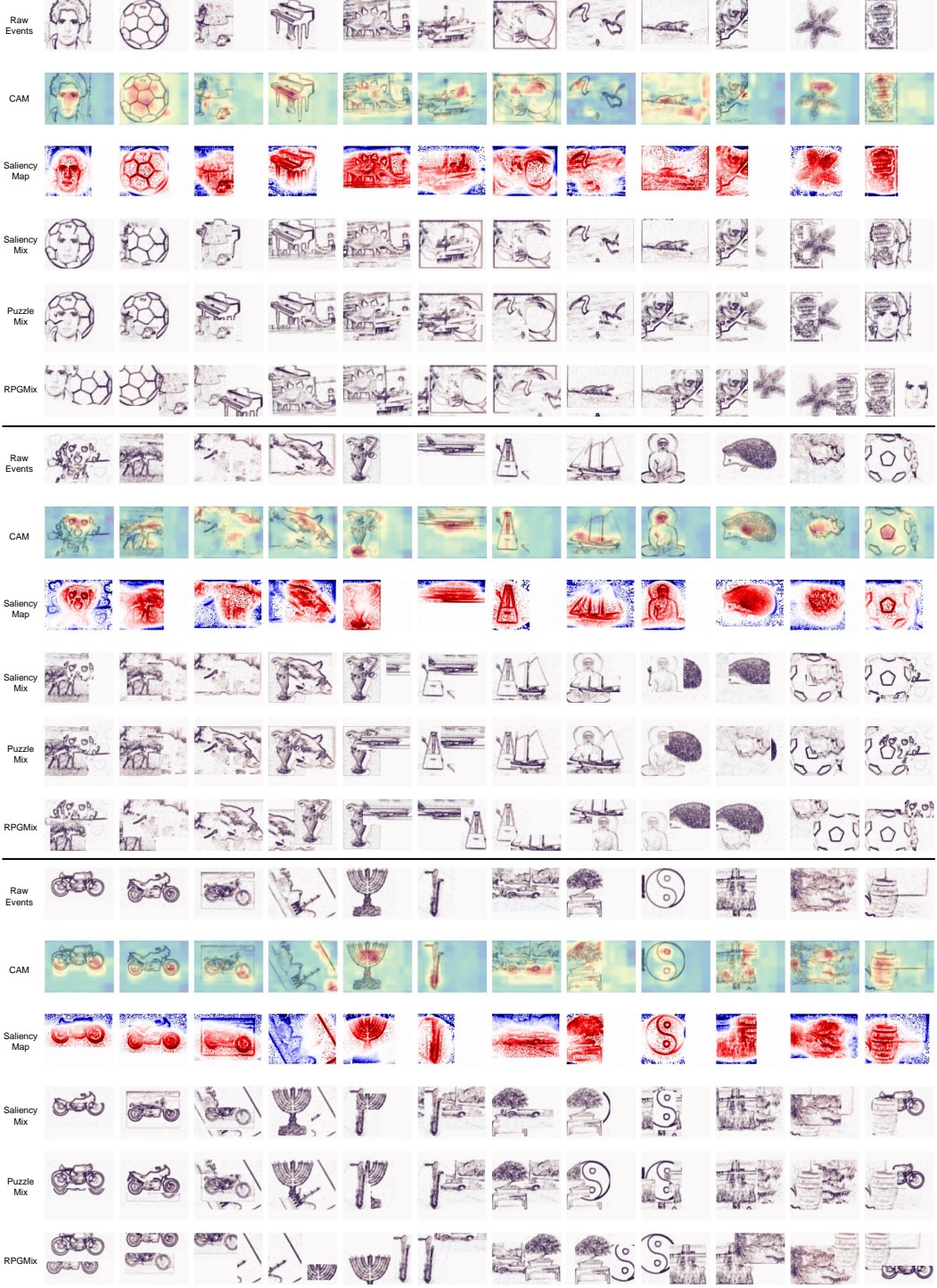

Figure 7: Mixed samples generated by different methods on N-Caltech101.

| Dataset | Model | Identity | Saliency Mix | Puzzle Mix | Puzzle Mix (mask only) | RPGMix |
|---|---|---|---|---|---|---|
| N-Caltech101 | Spike-VGG11 | 75.70 | 72.63 | 79.38 | 78.13 | **81.75** |
| SL-Animals-4Sets | SEW Resnet18 | 85.42 | 84.00 | 85.34 | 85.68 | **88.67** |
| SL-Animals-3Sets | SEW Resnet18 | 89.09 | 89.29 | 88.39 | 90.18 | **90.45** |

Table 6: Accuracy of Puzzle Mix, Saliency Mix, and RPGMix on object and action recognition tasks.

### A.3 ADDITIONAL QUAILITATIVE RESULTS

We present additional qualitative results of our method applied to the N-Caltech101, SL-Animals, and DVS-Gesture datasets. The saliency maps created using `SLTRP`, as shown in fig. 8, exhibit strong selectivity. They assign high values to label-related objects and low values to label-unrelated objects. In action recognition datasets, saliency maps primarily focus on label-related moving objects to capture the dynamic aspects of actions, as illustrated in fig. 9. In SL-Animals datasets (second row), the saliency maps focus on the person's hand raised to his/her head, assigning minimal attention to the lower area of the person, despite its high event density. Similarly, in the fourth row, the saliency maps show little interest in the person's right hand (from our perspective), focusing instead on the left hand, even though both hands have similar event densities. These results illustrate that it's the label-related actions, rather than event density, that truly draw the focus of the saliency maps. This observation further confirms the selectivity of SLTRP-generated saliency maps in both action recognition and object recognition datasets.

## B RELATED WORK

### B.1 EVENT-BASED DATA

Event-based data is generated by asynchronous sensors, usually referred to as event cameras (Mahowald, 1994; Son et al., 2017). Similar to point cloud, event-based data consists of four-dimensional points, denoted as $(x, y, t, p)$, where x and y are the spatial coordinates, t is the timestamp, and p is the polarity. Due to the inherent advantages of event cameras, event-based data is widely used in the field of perception, e.g., optical flow estimation (Gehrig et al., 2021), depth estimation (Wang et al., 2021), 3d reconstruction (Baudron et al., 2020; Rudnev et al., 2022), motion segmentation (Stoffregen et al., 2019; Zhou et al., 2021), semantic segmentation (Kim et al., 2022), etc.

There are two mainstream approaches to processing event-based data, frame-based approaches and event-by-event-based approaches. Frame-based approaches (Kogler et al., 2009; Bardow et al., 2016; Lagorce et al., 2016; Gehrig et al., 2019) are similar to image processing, where events are first converted into frames with a fixed shape $(C, H, W)$, and then the frames are fed into an ANN for downstream classification or regression tasks. Event-by-event-based approaches handle events on a one-by-one basis, which is natural for SNNs (Lee et al., 2020; Fang et al., 2021a; Deng et al., 2022).

### B.2 DATA AUGMENTATION

Vanilla data augmentations include simple transformations such as scale, rotation, flip, etc. Mixup (Zhang et al., 2018) proposes mixing two randomly selected samples and their corresponding labels as new data and label for training, based on which several works that modify the mixing details were proposed to further improve the robustness and performance (Yun et al., 2019; Hendrycks et al., 2019). Kim et al. (2020); Uddin et al. (2020) are similar to our proposed RPGMix that leverages the model's saliency information to augment the data, while our motivation and implementation details are quite different from them.

### B.3 RELEVANCE PROPAGATION

Relevance propagation was initially presented in (Montavon et al., 2017) as a method for creating saliency maps to visualize the impact of each pixel in the input data on the prediction of the model

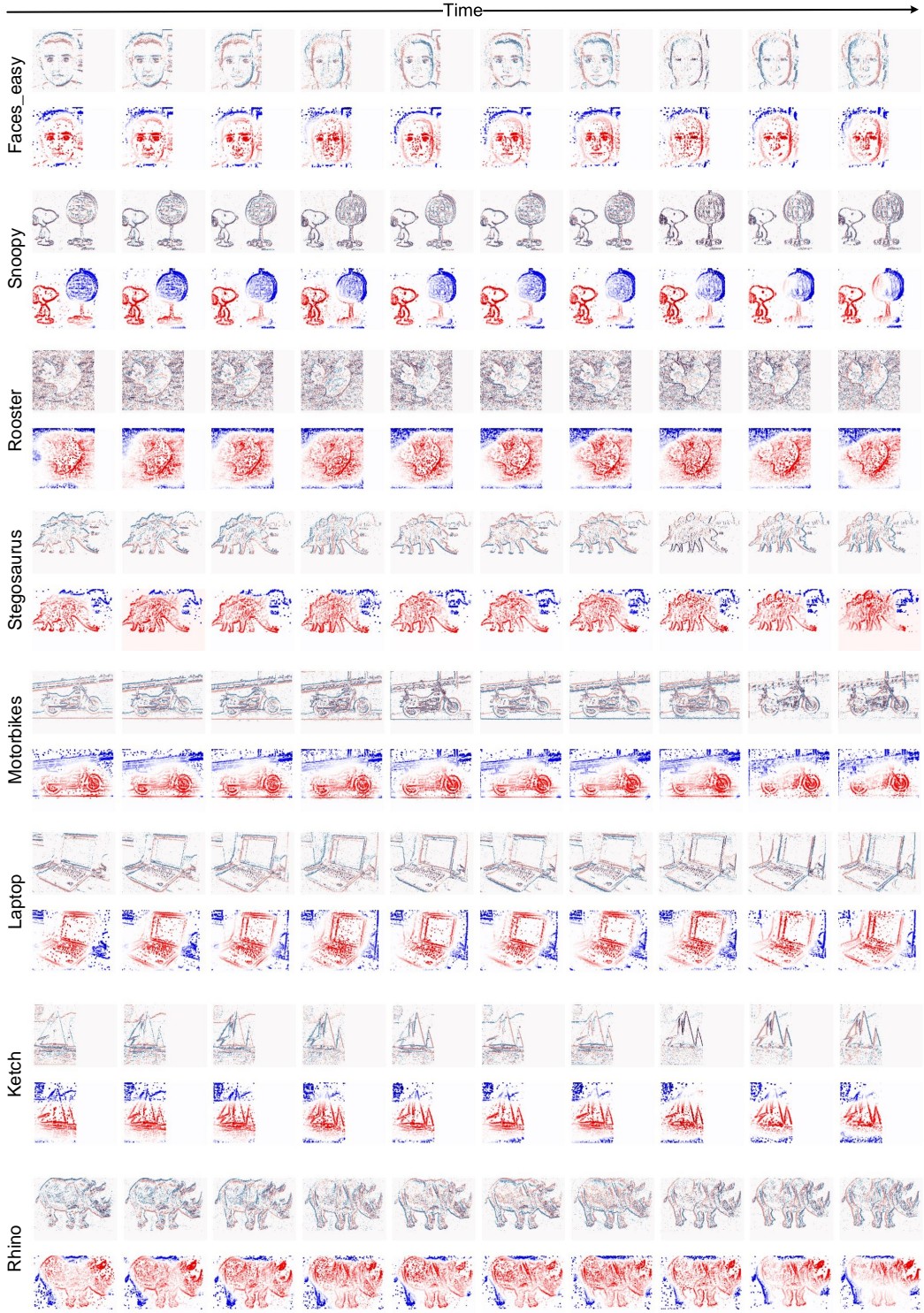

Figure 8: SLTRP-Saliency Maps from Spike-VGG11 on N-Caltech101.

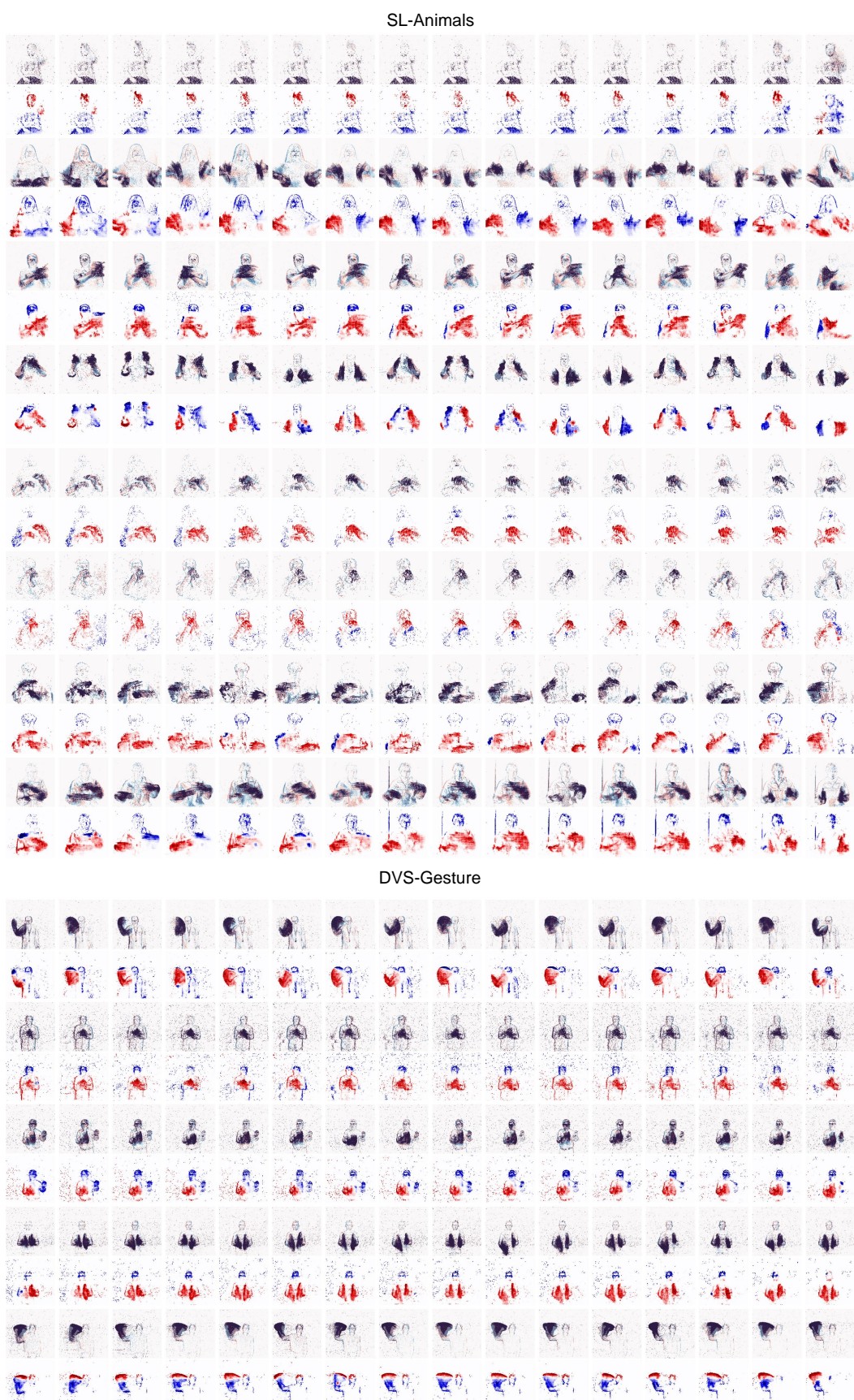

Figure 9: SLTRP-Saliency Maps from SEW Resnet-18 on SL-Animals and DVS-Gesture.

or a particular class, thereby enhancing the interpretability of neural networks. The specificity of saliency maps towards the target class can be improved by employing Contrastive Layer-wise Tolerance Propagation (CLRP) as described by Gu et al. (2018), which diminishes the relevance scores for non-target classes.

### B.4 CLASS ACTIVATION MAP

As another visualization tool describing the area of most interest to the model, Class Activation Map (CAM) is widely used to interpret the model's attention and to find objects belonging to a certain class in the input data. The original CAM (Zhou et al., 2016) could only visualize the activation map of the last global average pooling (GAP) layer of the model, requiring the model to have a special structure. Grad-CAM (Selvaraju et al., 2017) overcomes this limitation by replacing the weights of GAP with the sum of gradients, enabling the acquisition of CAM from any CNN model. On this basis, Grad-CAM++ (Chattopadhay et al., 2018), Score-CAM (Wang et al., 2020), and Relevance-CAM (Lee et al., 2021) were proposed to improve the quality of CAMs.

## C DATASETS AND TRAINING DETAILS

### C.1 OBJECT RECOGNITION TASK

**N-Caltech101** The neuromorphic version (Orchard et al., 2015) of Caltech101 (Fei-Fei et al., 2004). It is artificially created by moving an asynchronous time-based image sensor (ATIS) mounted on a pan tilt unit in front of an LCD screen that presents the image data in Caltech101. We use the same dataset as Li et al. (2022) that is split into the training set and test set by $9 : 1$. The resolution of N-Caltech101 is $180 \times 240$ which, in our implementation, is padded to $240 \times 240$ and rescaled to $128 \times 128$.

**CIFAR10-DVS** The DVS version (Li et al., 2017) of CIFAR10 (Krizhevsky et al., 2009). The generation of CIFAR10-DVS is similar to N-Caltech101. Following NDA (Li et al., 2022), we divide it by $9 : 1$ as the training set and test set, and scale the resolution from $128 \times 128$ to $48 \times 48$.

**N-Cars** A binary classification dataset (Sironi et al., 2018). Unlike N-Caltech101 and CIFAR10-DVS transformed from image datasets, N-Cars is obtained from the recording of an ATIS in real driving scenarios. Similar to the preprocessing of N-Caltech101, we first pad the resolution from $100 \times 120$ to $120 \times 120$ and then scale it to $48 \times 48$ on SNN.

**Mini N-ImageNet** A subset of N-ImageNet dataset (Kim et al., 2021). Despite being a smaller segment, it includes over 100,000 samples spanning 100 classes, making it the largest dataset used in our experiments. In line with the original authors' setup, we pad the resolution from $640 \times 480$ to $640 \times 640$ and then scale it down to $224 \times 224$ as the input resolution of the SNN.

### C.2 ACTION RECOGNITION TASK

For object recognition tasks, shapes and textures are the most important information to recognize an object, and thus we do not care how an object moves in the event streams. While in action recognition tasks, the movements of the objects should be considered. They represent two different recognition strategies and thus are both essential to validate our proposed method.

**DVS128 Gesture** A hand gesture dataset (Amir et al., 2017) recorded from a DVS128 camera. It comprises 1464 samples with 11 classes, split into the training set and test set by 8:2. The resolution is set as $128 \times 128$ on SNN.

**SL-Animals-DVS** A sign language dataset (Vasudevan et al., 2021) including 19 Spanish Sign Language signs corresponding to animals. Following the raw paper, we split it by 7.5:2.5 into the training set and test set. Also, we separate the datasets into "4 sets" and "3 sets", which excludes the samples disturbed by the indoor lighting conditions reflecting on the patterned clothing of the user. The resolution processing of SL-Animals-DVS is identical to that of DVS 128 Gesture.

| Neural Networks | Neuron Model | Datasets | Epoch | Batch Size | T | Learning Rate |
|---|---|---|---|---|---|---|
| Spike-VGG11 | LIF | N-Caltech101 CIFAR10-DVS N-Cars | 100 | 16 64 64 | 10 | $1 \times 10^{-3}$ |
| SEW Resnet-18 | PLIF | SL-Animals DVS-Gesture | 200 | 20 | 16 | $5 \times 10^{-4}$ |
| | | Mini N-Imagenet | 100 | 64 | 10 | $1 \times 10^{-3}$ |

Table 7: Hyper-parameters for different models and datasets.

### C.3 TRAINING DETAILS

In all experiments, we use Adam optimizer with the default setting $(\beta_1, \beta_2) = (0.9, 0.999)$ to update the parameters and Cosine Annealing Scheduler for the decrease of learning rate. Other hyper-parameters are shown in table 7. For fair comparisons, an identical seed is leveraged for all experiments. We utilize the public code of TET (Deng et al., 2022) to build Spike-VGG11 and Spikingjelly (Fang et al., 2020) to build SEW Resnet-18 with PLIF (Fang et al., 2021b).

## D PROOFS

In this section, we provide proofs of the propositions.

### D.1 PROOF FOR CONSERVATION PROPERTY ON $\alpha\beta$-RULE

*Proof.* The sum of relevance scores in layer $l-1$ is

$$
\begin{aligned}
\sum_i R_i^{(l-1)} = \sum_i \sum_j R_{i \leftarrow j}^{(l-1,l)} &= \sum_i \sum_j R_j^{(l)} \cdot \left( \alpha \cdot \frac{z_{ij}^+}{\sum_i z_{ij}^+} + \beta \cdot \frac{z_{ij}^-}{\sum_i z_{ij}^-} \right) \\
&= \sum_j R_j^{(l)} \sum_i \left( \alpha \cdot \frac{z_{ij}^+}{\sum_i z_{ij}^+} + \beta \cdot \frac{z_{ij}^-}{\sum_i z_{ij}^-} \right) \\
&= \sum_j R_j^{(l)} \left( \alpha \cdot \frac{\sum_i z_{ij}^+}{\sum_i z_{ij}^+} + \beta \cdot \frac{\sum_i z_{ij}^-}{\sum_i z_{ij}^-} \right) \\
&= \sum_j R_j^{(l)} (\alpha + \beta) \\
&= \sum_j R_j^{(l)}.
\end{aligned}
$$

$\square$

### D.2 PROOF FOR PROPOSITION 1

*Proof.* We prove proposition 1 by induction.

For case $t = 1$, according to eq. (12) we have

$$
R^{(l-1)}[1] = (1 - \gamma[1]) \left( \sum_{i=2}^{T} R^{(l)}[i] \prod_{j=2}^{i} \gamma[j] + R^{(l)}[1] \right),
$$

where $\gamma[1] = 0$, since at the first time step, the membrane voltage is initialized to be 0. Clearly, proposition 1 holds for case $t = 1$.

Next, suppose proposition 1 holds for case $t = k - 1$, we have

$$
\begin{aligned}
\sum_{t=1}^{k-1} R^{(l-1)}[t] &= \sum_{t=1}^{k-1} R^{(l)}[t] + \sum_{i=k}^{T} R^{(l)}[i] \prod_{j=k}^{i} \gamma[j] \\
&= \sum_{t=1}^{k-1} R^{(l)}[t] + \gamma[k] \left( R^{(l)}[k] + \sum_{i=k+1}^{T} R^{(l)}[i] \prod_{j=k+1}^{i} \gamma[j] \right) \\
&= \sum_{t=1}^{k-1} R^{(l)}[t] + \frac{\gamma[k]}{1 - \gamma[k]} R^{(l-1)}[k].
\end{aligned}
$$

Then

$$
\begin{aligned}
\sum_{t=1}^{k} R^{(l-1)}[t] &= \sum_{t=1}^{k-1} R^{(l-1)}[t] + R^{(l-1)}[k] = \sum_{t=1}^{k-1} R^{(l)}[t] + \frac{\gamma[k]}{1 - \gamma[k]} R^{(l-1)}[k] + R^{(l-1)}[k] \\
&= \sum_{t=1}^{k-1} R^{(l)}[t] + \frac{R^{(l-1)}[k]}{1 - \gamma[k]} \\
&= \sum_{t=1}^{k-1} R^{(l)}[t] + R^{(l)}[k] + \sum_{i=k+1}^{T} R^{(l)}[i] \prod_{j=k+1}^{i} \gamma[j] \\
&= \sum_{t=1}^{k} R^{(l)}[t] + \sum_{i=k+1}^{T} R^{(l)}[i] \prod_{j=k+1}^{i} \gamma[j].
\end{aligned}
$$

$\square$

