# OpenReview forum: "EventRPG: Event Data Augmentation with Relevance Propagation Guidance"
_ICLR.cc/2024/Conference — ICLR 2024 poster_

### Official Review · Reviewer_rLwX · 2023-10-14

**Soundness:** 3 good
**Presentation:** 3 good
**Contribution:** 2 fair
**Rating:** 6
**Confidence:** 3

**Summary:**

This paper proposes a mixup-based data augmentation method for training Spiking Neural Networks (SNNs). Inspired by saliency-based augmentation in RGB image vision such as Puzzle Mix and SaliencyMix, the authors derive the Class Activation Map (CAM) and saliency map for SNNs, and then mix two samples based on them. Experimental results show that the proposed EventRPG consistently improves the classification accuracy on various datasets.

**Strengths:**

- The derivation of the CAM and saliency map of SNNs itself is a clear contribution. The results in Table 1 prove the correctness of this
- EventRPG is able to consistently improve performance across event datasets
- The time cost of EventRPG is comparable to similar augmentations in conventional vision

**Weaknesses:**

My main concern is regarding the experiments:
- The paper motivates the need for event data augmentation with the statement that "the lack of huge event-based datasets similar to Imagenet prevents us from improving the model performance on relatively small datasets using a Pretrain-Finetune paradigm". However, there is an event camera version of ImageNet available [1], and its paper shows that pre-training on N-ImageNet can greatly improve the accuracy on other datasets via transfer learning. Therefore, this statement in the Introduction is wrong
- Since N-ImageNet is available, I would like to see results on this dataset. This is similar to conventional vision research on data augmentation, where ImageNet is the best testbed. If the authors are not able to train on N-ImageNet, the mini subset can be considered, though I do not think that is a comprehensive benchmark
- The authors compare EventRPG with conventional vision methods such as Grad-CAM in Table 1. Thus, I wonder if it is possible to compare with Puzzle Mix and SaliencyMix in Table 3?
- NDA applies the method to unsupervised contrastive learning and shows promising results. Is it possible to conduct such experiments using EventRPG?

[1] Kim, Junho, et al. "N-ImageNet: Towards robust, fine-grained object recognition with event cameras." ICCV. 2021.

**Questions:**

See Weaknesses

---

> ### Author Response · Authors · 2023-11-20
>
> > **Weakness 1**
> The paper motivates the need for event data augmentation with the statement that "the lack of huge event-based datasets similar to Imagenet prevents us from improving the model performance on relatively small datasets using a Pretrain-Finetune paradigm". However, there is an event camera version of ImageNet available [1], and its paper shows that pre-training on N-ImageNet can greatly improve the accuracy on other datasets via transfer learning. Therefore, this statement in the Introduction is wrong
>
> Thanks for pointing out this mistake. We've removed the misleading sentences accordingly in the revised version. Here's the revised content:
>
> "In terms of classification tasks, a number of event-based datasets, such as N-MNIST, N-Caltech101 [1], and CIFAR10-DVS [2], have been used to evaluate the performance of artificial neural networks (ANNs) and SNNs. **However, the issue of overfitting still poses a significant challenge for event-based datasets.** Data augmentation is an efficient method for improving the generalization and performance of a model."
>
> [1] Orchard, Garrick, et al. "Converting static image datasets to spiking neuromorphic datasets using saccades." Frontiers in neuroscience 9 (2015): 437.
>
> [2] Li, Hongmin, et al. "Cifar10-dvs: an event-stream dataset for object classification." Frontiers in neuroscience 11 (2017): 309.

---

> ### Author Response · Authors · 2023-11-20
>
> > **Weakness 2**
> Since N-ImageNet is available, I would like to see results on this dataset. This is similar to conventional vision research on data augmentation, where ImageNet is the best testbed. If the authors are not able to train on N-ImageNet, the mini subset can be considered, though I do not think that is a comprehensive benchmark
>
> Thanks for your advice. Mini N-ImageNet is a comprehensive datasets suitable for evaluating our method. We conduct experiments on this dataset and present them in the Appendix A.1. On this relatively difficult datasets, our method still achieves best results and is also shown to effectively alleviate the overfitting problem. As shown by the notable positive effects, augmentations significantly enhance the performance of the model. In this comparison, our approach outperforms the nearest competitor NDA, achieving a higher Top-1 accuracy by $5.06\\%$ and Top-5 accuracy by $4.1\\%$. Also the loss curves in Fig. 6 highlight the efficiency of our approach in boosting the model's performance and mitigating overfitting.
>
> | Data Augmentation | Identity | EventDrop | NDA    | EventRPG |
> | :------------------ | :------- | :-------- | :----- | :------- |
> | Top-1 Accuracy      | 28\.16   | 34\.18    | 35\.84 | **40\.90**   |
> | Top-5 Accuracy      | 52\.14   | 60\.94    | 63\.64 | **67\.74**   |

---

> ### Author Response · Authors · 2023-11-20
>
> > **Weakness 3**
> The authors compare EventRPG with conventional vision methods such as Grad-CAM in Table 1. Thus, I wonder if it is possible to compare with Puzzle Mix and SaliencyMix in Table 3?
>
> Thanks for your comments. Comparing our RPGMix method with Puzzle Mix and SaliencyMix is crucial for a thorough analysis. We compared various saliency-based mixing methods on N-Caltech101 and SL-Animals datasets, with results detailed in Appendix A.2.
>
> | Dataset          | Model        | Identity | Saliency Mix | PuzzleMix | PuzzleMix (mask only) | RPGMix |
> | :--------------- | :----------- | -------: | -----------: | ---------------------------: | ---------------------------: | -----: |
> | N-Caltech101     | Spiking-VGG11       | 75\.7    | 72\.63       | 79\.38                       | 78\.13                       | **81\.75** |
> | SL-Animals-4Sets | SEW-Resnet18 | 85\.42   | 84           | 85\.34                       | 85\.68                       | **88\.67** |
> | SL-Animals-3Sets | SEW-Resnet18 | 89\.09   | 89\.29       | 88\.39                       | 90\.18                       | **90\.45** |
>
> Contrary to Saliency Mix and Puzzle Mix, which can sometimes reduce performance, our RPGMix consistently has a positive impact on both datasets. This proves the advantage of RPGMix compared with Puzzle Mix and Saliency Mix on event-based data.

---

> ### Author Response · Authors · 2023-11-20
>
> > **Weakness 4**
> NDA applies the method to unsupervised contrastive learning and shows promising results. Is it possible to conduct such experiments using EventRPG?
>
> In line with NDA, we apply SimSiam [1] for unsupervised pretraining on a Resnet-18 model, distinguishing between non-mixing and mixing augmentations. SimSiam employs non-mixing augmentations to create varied samples from a single one, which we adopt for EventDrop's unsupervised pretraining, as it exclusively uses non-mixing augmentations. For EventRPG and NDA, which incorporate both types of augmentations, we follow the Mixsiam [2] approach for unsupervised contrastive pretraining.
>
> Within the SimSiam training framework, the predictor's output layer does not reflect classification results. Therefore, implementing CLRP at the output layer is not feasible. To address this, we initialize the relevance scores of the output layer with the neural network's output values.
>
> During the finetuning phase, to ensure a fair comparison, we do not apply any augmentations to the data. As a result, our outcomes are expected to be lower than those presented in the original paper by NDA [3].
>
> | Radom Initialized | ImageNet Pretrain | EventDrop | NDA    | EventRPG |
> | :---------------- | :---------------- | :-------- | :----- | :------- |
> | 47\.56            | 74\.32            | 74\.56    | 74\.56 | **76\.81**   |
>
> The table shows that pretraining markedly improves model performance. Models pretrained using EventDrop and NDA perform comparably to those pretrained on ImageNet. However, models pretrained with our approach show a notable $2.49\\%$ improvement. This highlights the promise of applying EventRPG to unsupervised contrastive learning, an area we intend to investigate more in future work. Therefore, these results are not included in the current manuscript.
>
> [1] Chen, Xinlei, and Kaiming He. "Exploring simple siamese representation learning." In CVPR 2021.
>
> [2] Guo, Xiaoyang, et al. "Mixsiam: a mixture-based approach to self-supervised representation learning." arXiv preprint arXiv:2111.02679 (2021).
>
> [3] Li, Yuhang, et al. "Neuromorphic data augmentation for training spiking neural networks." In ECCV 2022.

---

> > ### Comment · Reviewer_rLwX · 2023-11-20
> > **Re: Rebuttal**
> >
> > I thank the authors for the detailed replies and additional experiments. Can you explain more about the last point, especially the statement:
> >
> > > During the finetuning phase, to ensure a fair comparison, we do not apply any augmentations to the data. As a result, our outcomes are expected to be lower than those presented in the original paper by NDA [3].
> >
> > Will you achieve better results and outperform NDA if you augment the training data? I do not see any issues doing this (I assume this is also what NDA does). Maybe one issue is about the training speed/computation overhead brought by the data augmentation.
> > Thus one more question I have is how is the speed of your method compared to NDA?

---

> ### Author Response · Authors · 2023-11-21
>
> Our expression "During the finetuning phase, to ensure a fair comparison, we do not apply any augmentations to the data. As a result, our outcomes are expected to be lower than those presented in the original paper by NDA [3]." may causes misunderstanding. Here we put more details and hope it will be clearer:
>
>
> In our experiments on contrastive learning, we pretrain the model using NDA, EventDrop, and EventRPG, respectively, while no augmentation is used in the finetuning stage. If we augment the data in the finetuning stage, we would not know how much improvement the pretraining brings, since the data augmentation in the pretraining stage and the finetuning stage both bring improvements to the performance. Excluding the data augmentation in the finetuning stage is helpful for a fair comparison. Under this setting, our method is better than NDA.
>
>
>
> If we leverage data augmentation in both pretraining and finetuning stage as what NDA does, EventRPG still achieves better results with an improvement of $0.78\\%$ compared to NDA, as shown in the table below (The result of NDA under this setting is from its original paper [1]). We hope you are satisfied with the experiment.
>
>
> | Pretrain Augmentation | Finetune Augmentation | Pretrain Epoch | Finetune Epoch | Accuracy   |
> | :---------------------: | :-------------------: | :------------: | :------------: | :--------: |
> | NDA                     | Identity              | 600            | 100            | 74\.56     |
> | EventRPG                     | Identity              | 600            | 100            | **76\.81** |
> | NDA                     | NDA                   | 600            | 100            | 80\.80      |
> | EventRPG                     | EventRPG                   | 600            | 100            | **81\.58** |
>
> We believe that EventRPG is the superior choice, both during the pretraining and finetuning stages, as demonstrated by the experimental results.
>
> We also record the time cost durint the finetuning, and the results are shown below. The time cost of EventRPG is very close to that of NDA, indicating that our method is time-efficient for ANNs.
>
>
> |                                     | EventRPG | NDA   |
> | :----------------------------------: | :--------: | :----: |
> | Augmentation Time (ms/event stream) | 6\.17     | 5\.43 |
>
>
> Additionally, We carried out experiments comparing the time costs (ms per event stream) of different augmentation methods on SNN at CIFAR10DVS dataset, as shown in the table below.
>
> | Batch Size per GPU | EventDrop | NDA      | EventRPG       |
> | :----------------: | :-------: | :------: | :-------: |
> | 1                  | 5\.00743  | 5\.29057 | 11\.05086 |
> | 2                  | 5\.2048   | 5\.63054 | 7\.83654  |
> | 4                  | 5\.24821  | 5\.95036 | 6\.46512  |
> | 8                  | 4\.88     | 5\.9433  | 6\.13769  |
> | 16                 | 4\.93906  | 6\.14297 | 5\.55679  |
> | 32                 | 4\.65249  | 6\.03272 | 5\.34178  |
>
> It can be seen that when the batch size of each GPU surpasses 4, the time cost of EventRPG is close to those of EventDrop and NDA. More details can be found in the Section 5.2.3 of our revised paper.
>
> We thank you again for your suggestions, which have been very helpful in improving our paper. Please let us know if you have further questions.
>
> [1] Li, Yuhang, et al. "Neuromorphic data augmentation for training spiking neural networks." In ECCV 2022.

---

> > ### Comment · Reviewer_rLwX · 2023-11-21
> > **Re: Rebuttal**
> >
> > Thanks. I am satisfied with the results and have updated my score.

---

> > > ### Author Response · Authors · 2023-11-22
> > >
> > > We are delighted to see that the major concerns raised by the reviewer have been successfully addressed. We deeply appreciate the reviewer's dedicated time and effort.

---

### Official Review · Reviewer_DHWr · 2023-10-31

**Soundness:** 3 good
**Presentation:** 2 fair
**Contribution:** 3 good
**Rating:** 5
**Confidence:** 3

**Summary:**

This paper proposes two efficient and practical methods, SLTRP and SLRP, for generating CAMs and saliency maps for SNNs for the first time. Based on these, the authors propose EventRPG to achieve data augmentation, which drops events and mixes events with Relevance Propagation Guidance.

**Strengths:**

This paper proposes Spiking Layer-wise Relevance Propagation(SLRP) rule and Spiking Layer-Time-wise Relevance Propagation(SLTRP) rule, the layer-wise relevance propagation method of SNNs for the first time, which can obtain the feature contribution at each pixel. RGBDrop and RGBMix are established to achieve data augmentation based on the generated CAMs. The results of experiments prove the usefulness of SLRP and SLTRP both for accuracy and efficiency. The EventRPG shows good performance in object recognition and action recognition tasks.

**Weaknesses:**

1. The description of RPGMix is quite simple and unclear. Section 4.3 fails to clearly illustrate the algorithm flow of RPGMix. Many operations in Fig 3(b) are not carefully analyzed, such as sample position in the Nonoverlapping Region, which makes it hard to understand. I think Fig3 is the main figure of this paper and Fig 3(b) accounts for the most part of Fig3, hence the authors need to spend more space to describe it. Otherwise, the readers may feel confused about RPGMix.
2. Equation 15 lacks physical meaning and theoretical basis. It needs more explanations for researchers to make further progress.
3. The objective faithfulness of SLRP and SLTRP is not outstanding enough. In N-Cars,  DVSGesture, and SL-Animals datasets, the proposed methods have similar or even inferior performance compared to other algorithms.

**Questions:**

The author should carefully illustrate RPGMix and the performances of SLRP and SLTRP are challenged in the metric of the objective faithfulness.

**Details Of Ethics Concerns:**

No ethics concerns.

---

> ### Author Response · Authors · 2023-11-20
>
> > **Weakness 1**
> The description of RPGMix is quite simple and unclear. Section 4.3 fails to clearly illustrate the algorithm flow of RPGMix. Many operations in Fig 3(b) are not carefully analyzed, such as sample position in the Nonoverlapping Region, which makes it hard to understand. I think Fig3 is the main figure of this paper and Fig 3(b) accounts for the most part of Fig3, hence the authors need to spend more space to describe it. Otherwise, the readers may feel confused about RPGMix.
>
> > **Weakness 2**
> Equation 15 lacks physical meaning and theoretical basis. It needs more explanations for researchers to make further progress.
>
> We apologize for the earlier lack of clarity. We have added the motivation and more details of the implementation in the Section 4.3. Here's the content:
>
> "**Event-based data, in contrast to image-based data, does not include color details, with the most crucial aspect being the texture information it contains. The overlapping of label-related objects will impair the texture details of these objects, which in turn further degrades the quality of features extracted in SNNs. Building upon this motivation, we propose Relevance Propagation Guided Event Mix (RPGMix).** The whole mixing strategy is illustrated in Fig. 2b. For two event-based data candidates, we utilize relevance propagation to localize the label-related regions and obtain two bounding boxes. **To mix two objects with clear texture features, we randomly select two positions ensuring minimal overlap of their bounding boxes. This involves initially positioning one box at a corner to maximize the nonoverlapping area for the other box's placement, then selecting positions for both boxes in order, maintaining minimal overlap and maximizing sampling options.** Finally, the two event streams are moved to the sampled positions. Although this linear translation part prevents the overlapping of label-related objects, the background of one object would still overlap with the other object. Moreover, in one single time step, the representation ability of the spiking neurons (which only output binary information) is much worse than that of the activation layer (usually ReLU) of ANNs, making them less capable of spatial resolution and more likely to fall into local optima. Therefore, to promise the presence of only events from a single event stream candidate per pixel, avoiding regions with mixed information from interfering with the SNN, we adopt a CutMix strategy to mask the two event streams based on the bounding box of the second event stream, as demonstrated in the left part in Fig. 2b. **[1] takes the sum of each sample's mask as the ratio of their corresponding labels. This ensures that the proportion of labels in the mixed label matches the proportion of pixels belonging to each sample. In our approach, we further aim to align the proportion of labels with the proportion of label-related pixels, which can be estimated using the bounding boxes. As a result, the labels of the two event streams are mixed as**
> $$L_{mix} = \frac{L_{1}(w_{1}h_{1} - S_{overlap}) + L_{2}w_{2}h_{2}}{w_{1}h_{1} + w_{2}h_{2} - S_{overlap}},$$
> where $w_{i}$ and $h_{i}$ denote the width and height of the object in the event stream $i$. $L_1$ and $L_2$ are the one-hot labels of the two event streams and $S_{overlap}$ is the area of the overlapping region of the two bounding boxes."
>
>
>
> [1] Kim, Jang-Hyun, Wonho Choo, and Hyun Oh Song. "Puzzle mix: Exploiting saliency and local statistics for optimal mixup." In ICML 2020.

---

> ### Author Response · Authors · 2023-11-20
>
> > **Weakness 3**
> The objective faithfulness of SLRP and SLTRP is not outstanding enough. In N-Cars, DVSGesture, and SL-Animals datasets, the proposed methods have similar or even inferior performance compared to other algorithms.
>
> In this comparison, although traditional methods may perform well on specific datasets, our methods stand out for their overall superior performance. To provide a comprehensive comparison, we calculated the average improvement percentage for each method.
>
> The A.I. or A.D. score of method $ i $ on dataset $j $ is denoted as $ v_{i, j} $. We compute each dataset's mean value as $ v_{mean,j}=\\frac{1}{N_{a}}\\sum_i v_{i,j} $, where $ N_a $ represents the count of augmentations. The augmentation improvement percentage is $ v'\_{i,j} =\\frac{(v_{i, j} - v_{mean, j})}{v_{mean, j}}$. Finally, we average the improvement for each method across datasets for A.I. or A.D. as $ I_{i}=\\frac{1}{N_{d}}\\sum_j v'_{i,j} $, with $ N_d $ being the number of datasets. Here's the averaged improvement of A.I. and A.D. In this comprehensive statistical analysis, SLTRP-Saliency Map and SLRP-Saliency Map surpass the performance of other methods, proving their efficiency.
>
> | Method             | Average A.I. Improvement ↑ | Average A.D. Improvement ↓ |
> | :----------------- | -------------------------: | -------------------------: |
> | SAM                | -0\.52                     | -0\.04                     |
> | Grad-CAM           | -0\.43                     | 1\.29                      |
> | Grad-CAM++         | -0\.07                     | -0\.30                     |
> | SLRP-RelCAM        | 0\.13                      | -0\.25                     |
> | SLTRP-RelCAM       | 0\.13                      | -0\.25                     |
> | SLRP-CAM           | -0\.05                     | 0\.18                      |
> | SLTRP-CAM          | -0\.05                     | 0\.18                      |
> | SLRP-Saliency Map  | **0\.44**                      | **-0\.41**                     |
> | SLTRP-Saliency Map | 0\.42                      | -0\.40                     |                 |
>
> Additionally, CAMs produced by previous methods highlight the model's focused regions at the feature map level, while saliency maps obtained from SLRP and SLTRP disclose the model's attention at a more specific pixel level.

---

### Official Review · Reviewer_dKSN · 2023-11-04

**Soundness:** 4 excellent
**Presentation:** 4 excellent
**Contribution:** 3 good
**Rating:** 8
**Confidence:** 3

**Summary:**

The paper addresses the problem of event-based data augmentation for Spiking Neural Networks by computing saliency. Two methods are presented with different levels of overhead on training; these methods are incorporated in two augmoentation schemes that either drop events or mix two event streams. A number of SNNs was evaluated on both object and action recognition datasets, and improvement in accuracy was demonstrated - especially on action recognition datasets.

**Strengths:**

1) The authors mention that the code will be released; to me this is important as it enables other researchers to easily build on top of this paper.

2) I particularly enjoyed the theoretical introduction into the SNNs, this makes the rest of the paper much easier to read.

**Weaknesses:**

1) If anything, I would like to notice here that the improvements compared to competing methods are generally small. It would be interesting to see an apples-to-apples comparison in therms of compute overhead (which is mentioned, but I do not see numberical benchmark results); or, another compelling reason to use the presented methods vs e.g. second best.

2) Also see 'questions': it would be good to show that saliency on motion-related datasets is more than just event density. This could be done e.g. by correlating saliency to raw optical flow magnitude, or evaluating on actions that are slower.

**Questions:**

1) In abstract, expand SNN as 'Spiking Neural Networks', for the benefit of readers unfamiliar with the abbreviation.

2) Fig. 1 highlights a failure case of one of the augmentation methods; it would be great to see a citation that explains why this limitation is difficult to alleveiate (e.g. by considering correct label classes). The image also seems excessive, and a single-sentence explanation should be enough.

3) I am curious if the better performance / saliency on action-related datasets is an artifact of the event camera itself - since the density of the events correlates with motion and SNN may get more cues on regions with faster motion.

4) It would be beneficial, in tables 3 and 4 to mention that all of the datasets are 'native' event-based datasets (unless I am mistaken). What would be the comparison if a classic camera dataset was converted to events, especially on action classification tasks?

---

> ### Author Response · Authors · 2023-11-20
>
> > **Weakness 1** If anything, I would like to notice here that the improvements compared to competing methods are generally small. It would be interesting to see an apples-to-apples comparison in therms of compute overhead (which is mentioned, but I do not see numberical benchmark results); or, another compelling reason to use the presented methods vs e.g. second best.
>
> Thanks for pointing out this missing experiment. In the updated version of our paper, we've included time consumption experiments (refer to section 5.2.3). It can be seen that as batch size increases, the time cost of our method decreases in a near-linear manner. Particularly when the batch size per GPU exceeds 4, the augmentation time for EventRPG aligns closely with that of EventDrop and NDA. Additionally, we've conducted new experiments comparing various augmentation methods on the mini N-ImageNet dataset (see Appendix A.1) and in the context of unsupervised contrastive learning (refer to our responses to Reviewer rLwX's W4 query). Under these scenarios, our method surpasses both EventDrop and NDA without incurring extra computational costs, as all batch sizes per GPU are set above 4. Additionally, to the best of our knowledge, our results on N-Caltech101, CIFAR10-DVS, and SL-Animals represent the state-of-the-art (SOTA) in the direct training track for Spiking Neural Networks (SNNs).

---

> ### Author Response · Authors · 2023-11-20
>
> > **Weakness 2**  Also see 'questions': it would be good to show that saliency on motion-related datasets is more than just event density. This could be done e.g. by correlating saliency to raw optical flow magnitude, or evaluating on actions that are slower.
>
> > **Question 3** I am curious if the better performance / saliency on action-related datasets is an artifact of the event camera itself - since the density of the events correlates with motion and SNN may get more cues on regions with faster motion.
>
> Your suggestion to validate our method's selectivity via qualitative comparison is excellent. We attempted to replicate leading optical flow estimation works like TMA [1] and E-RAFT [2], but their primary training on autonomous driving datasets somewhat restricts their effectiveness in object recognition tasks. As an alternative, we have visualized additional saliency map results using SLTRP in Appendix A.3. We found that saliency maps don't concentrate on high event density areas, but rather on objects or actions related to the label. For instance, in SL-Animals results (second row), the saliency maps focus on the person's hand raised to their head, assigning minimal attention to the lower area of the person, despite its high event density. Similarly, in the fourth row, the saliency maps show little interest in the person's right hand (from our perspective), focusing instead on the left hand, even though both hands have similar event densities. We believe that it's the label-related actions, rather than event density, that truly draw the focus of the saliency maps. This observation further confirms the selectivity of the SLTRP-generated saliency maps in both action recognition and object recognition datasets (see Fig. 8).
>
> [1] Liu, Haotian, et al. "TMA: Temporal Motion Aggregation for Event-based Optical Flow." In ICCV 2023.
>
> [2] Gehrig, Mathias, et al. "E-raft: Dense optical flow from event cameras." In 3DV 2021.

---

> ### Author Response · Authors · 2023-11-20
>
> > **Question 1** In abstract, expand SNN as 'Spiking Neural Networks', for the benefit of readers unfamiliar with the abbreviation.
>
> > **Question 2** Fig. 1 highlights a failure case of one of the augmentation methods; it would be great to see a citation that explains why this limitation is difficult to alleveiate (e.g. by considering correct label classes). The image also seems excessive, and a single-sentence explanation should be enough.
>
> Thanks for the clarification. We expand "SNN" to "Spiking Neural Networks" in the Abstract and instead of Fig. 1, we include a brief citation to introduce the failure case, maintaining brevity in the text. This also helps to save more space for other valuable content.
>
> Here's the revised version of the introduction to failure cases:
>
> "Nevertheless, current mixing augmentation strategies in event-based field do not consider the size and location information of label-related objects, **and thus may produce events with incorrect labels and disrupt the training process [1, 2]. To address this problem in image processing field, They mix the label-related objects together based on the saliency information obtained from neural networks.** This paradigm achieves better results compared with conventional non-saliency augmentations."
>
> [1] Uddin, A. F. M., et al. "Saliencymix: A saliency guided data augmentation strategy for better regularization." In ICLR 2020.
>
> [2] Kim, Jang-Hyun, Wonho Choo, and Hyun Oh Song. "Puzzle mix: Exploiting saliency and local statistics for optimal mixup." In ICML 2020.

---

> ### Author Response · Authors · 2023-11-20
>
> > **Question 4** It would be beneficial, in tables 3 and 4 to mention that all of the datasets are 'native' event-based datasets (unless I am mistaken). What would be the comparison if a classic camera dataset was converted to events, especially on action classification tasks?
>
> In our experiments, we used three object recognition datasets. Two of them, N-Caltech101 and CIFAR10-DVS, are both derived by moving an event camera in front of static images from the Caltech-101 [1] and CIFAR-10 [2] image datasets, respectively. We've included explanations regarding this in the captions of Tables 3 and 4. Converting video frames into event data is indeed a promising area, as it can significantly diversify event-based datasets. However, the quality of events generated from videos through simulation largely depends on the video frame rate. A lower frame rate can lead to discontinuity in the time dimension of the converted events, thereby increasing the disparity between simulated and real event datasets. We recognize that pretraining on these simulated datasets and then fine-tuning on real event-based datasets is an intriguing approach. We plan to delve into this research area in the near future.
>
> [1] Fei-Fei, Li, Rob Fergus, and Pietro Perona. "Learning generative visual models from few training examples: An incremental bayesian approach tested on 101 object categories." In CVPRW 2004.
>
> [2] Krizhevsky, Alex, and Geoffrey Hinton. "Learning multiple layers of features from tiny images." (2009): 7.

---

### Author Response · Authors · 2023-11-20

We thank the reviewers for their insightful feedback and apologize for the delayed response, due to conducting extensive supplementary experiments and significant manuscript revisions. We're grateful for the recognition of our novel relevance propagation rules for SNNs and our saliency-based data augmentation methods from all reviewers. We've carefully addressed all comments, eliminating redundant content, clarifying ambiguities, and adding experiments to validate our method's effectiveness. Key updates in the revised manuscript are marked in orange:

1. **Time Consumption Evaluation**: Section 5.2.3 details our method's time efficiency compared to EventDrop and NDA. When batch size per GPU exceeds 4, EventRPG costs similar time as NDA and EventDrop.
2. **Mini N-ImageNet Experiments**: Appendix A.1 shows our approach's efficacy in enhancing model performance and reducing overfitting on a large dataset with over 100,000 samples across 100 classes.
3. **Saliency-Based Mixing Methods**: Appendix A.2 presents comparisons on N-Caltech101 and SL-Animals with Saliency Mix and Puzzle Mix, highlighting RPGMix's superior performance and robust generalization on event-based datasets.
4. **Qualitative Results**: Appendix A.3 displays CAMs, saliency maps, and mixed samples, illustrating the distinctions among methods and our method's capacity to maintain salient information.
5. **Unsupervised Learning Experiment**: As responded to Reviewer rLwX's W4, we demonstrate our method's potential in unsupervised learning, with models pretrained by EventRPG showing high adaptability to CIFAR10-DVS.

---

### Author Response · Authors · 2023-11-22

We sincerely appreciate the time and effort everyone has devoted to improving our work with your valuable suggestions. Your comments have notably enhanced its quality. We hope our responses have thoroughly addressed all your queries. Should you have any further questions or need clarifications, please feel free to reply. We are committed to responding promptly.

Just a gentle reminder, we are approaching the end of the discussion period. Your timely feedback would be greatly appreciated for the smooth progression of our submission.

---

### Meta-Review · Area_Chair_Vvjv · 2023-12-06

**Metareview:**

The paper introduces the means to compute saliency for spiking neural networks (SNN) with the goal of event-based data augmentation. As also acknowledged by the reviewers, this contribution is both meaningful and novel. In addition, reviewers complemented the nice theoretical introduction to SNNs, highlight that the derivation for the saliency is a clear and correct contribution, and agree that the provided empirical evidence supports the made claims. The initial weaknesses, resulting in borderline ratings, revolved in very large parts around further experimentation (in additional settings) and further comparison, with some lesser concerns on providing additional detail with respect to computation time and qualitative examples.

In the discussion and through a respective pdf revision, the authors have clarified the majority of these concerns. In particular, multiple sets of new experiments on new datasets, investigation of an unsupervised setting, comparison with further baselines, and time consumption evaluation have been provided. They seem to improve the support for the made claims and substantiate them with further empirical evidence. Correspondingly, the reviewers who engaged in discussion have updated and raised their score towards acceptance. The AC believes that overall the made improvements warrant publication of the paper, as they have addressed the main points that were necessary to be improved.

**Justification For Why Not Higher Score:**

Although improvements are demonstrated, many of the shown improvements remain fairly small. From an empirical standpoint, the paper thus presents a valid contribution, especially in light of the derived methodology itself. The paper should therefore be accepted, yet does not present enough of a breakthrough to be considered as a highlighted (oral) contribution at the conference.

**Justification For Why Not Lower Score:**

The raised primary weaknesses revolved in large parts around further experimentation, for which a substantial amount of further empirical evidence has been included in the revised paper.

---

### Decision · Program_Chairs · 2024-01-16

Accept (poster)